# Autophagy in Muscle Regeneration: Mechanisms, Targets, and Therapeutic Perspective

**DOI:** 10.3390/ijms252211901

**Published:** 2024-11-06

**Authors:** Yun Chu, Xinrun Yuan, Yiming Tao, Bin Yang, Jinlong Luo

**Affiliations:** 1Department of Intensive Care Medicine, Tongji Hospital, Tongji Medical College, Huazhong University of Science and Technology, Wuhan 430030, China; m202276297@hust.edu.cn (Y.C.); taoyiming525@163.com (Y.T.); d202282209@hust.edu.cn (B.Y.); 2Department of Emergency, Tongji Hospital, Tongji Medical College, Huazhong University of Science and Technology, Wuhan 430030, China; yuanxinrun@163.com

**Keywords:** autophagy, muscle regeneration, satellite cells, apoptosis, cell therapy

## Abstract

Autophagy maintains the stability of eukaryotic cells by degrading unwanted components and recycling nutrients and plays a pivotal role in muscle regeneration by regulating the quiescence, activation, and differentiation of satellite cells. Effective muscle regeneration is vital for maintaining muscle health and homeostasis. However, under certain disease conditions, such as aging, muscle regeneration can fail due to dysfunctional satellite cells. Dysregulated autophagy may limit satellite cell self-renewal, hinder differentiation, and increase susceptibility to apoptosis, thereby impeding muscle regeneration. This review explores the critical role of autophagy in muscle regeneration, emphasizing its interplay with apoptosis and recent advances in autophagy research related to diseases characterized by impaired muscle regeneration. Additionally, we discuss new approaches involving autophagy regulation to promote macrophage polarization, enhancing muscle regeneration. We suggest that utilizing cell therapy and biomaterials to modulate autophagy could be a promising strategy for supporting muscle regeneration. We hope that this review will provide new insights into the treatment of muscle diseases and promote muscle regeneration.

## 1. Introduction

Satellite cells, also known as muscle stem cells (MuSCs) when activated, are essential for muscle repair. These adult stem cells, unique to skeletal muscle, typically remain in a dormant state [1]. In response to tissue injury, these cells are rapidly activated and divided to generate new stem cells, which proliferate to form myoblasts, which further differentiate into myocytes to repair damaged muscle tissue [2,3]. However, muscle regeneration can be significantly impaired under various conditions due to dysfunctional satellite cell activity [3,4]. For instance, during aging, satellite cells lose their regenerative capacity and exhibit signs of senescence [5], while muscle atrophy caused by denervation leads to premature differentiation [6]. Therefore, restoring satellite cell function is vital for enhancing muscle regeneration.

Autophagy is essential for maintaining tissue and organ homeostasis and contributes significantly to tissue repair and regeneration. In the liver, disruption of autophagy can impair the regenerative capacity [7]. Autophagy regulates various metabolic and stress-related cellular signals that can affect satellite cells during quiescence, activation and differentiation [8,9], thereby influencing the muscle regeneration process.

Autophagy begins with the ULK (Unc-51-like kinase) complex, which triggers the formation of autophagosomes based on nutritional status and energy levels [10]. For example, AMP-activated protein kinase (AMPK) is activated under energy stress, while the mammalian target of rapamycin complex 1 (mTORC1) responds to nutrient availability [11,12]. Autophagy-related genes (ATGs) are essential for the formation of autophagosomes and their fusion with lysosomes, enabling the degradation and recycling of cellular components [13,14]. Moreover, ATG proteins influence various cellular processes, including the regulation of the cell cycle, with ATG7 linked to the expression of the cell cycle inhibitor CDKN1A [15,16].

Autophagy also interacts with apoptosis, influencing cell fate decisions, such as the survival or death of satellite cells, which are essential for muscle regeneration [17]. Disruptions in autophagy are associated with the onset and progression of various diseases, and therapeutic modulation of autophagy through pharmacological or molecular interventions has shown potential [18,19]. However, the exact molecular pathways by which autophagy affects muscle regeneration remain to be fully elucidated. This review explores recent research on the role of autophagy in muscle regeneration and discusses potential strategies for harnessing autophagy to improve regenerative outcomes.

## 2. The Process of Autophagy

Autophagy, a critical cellular degradation and recycling process, can be classified into three main types based on substrate delivery to lysosomes: microautophagy, chaperone-mediated autophagy, and the classical autophagy pathway [20]. The classical pathway progresses through several stages, including phagophore formation, nucleation, elongation, fusion of autophagosomes with lysosomes, and subsequent degradation [20]. AMPK activates ATG13 and unc-51-like autophagy-activating kinase 1 (ULK1) to form the ULK1 complex, which triggers autophagy [21]. This complex activates the VPS34 complex by incorporating phosphatidylinositol (PI) to facilitate phagophore formation. Vacuolar protein sorting 34 (Vps34) uses PI as a substrate to produce phosphatidylinositol 3-phosphate (PI3P). This crucial step is enhanced by the interaction between Beclin-1 and Vps34, which increases PI3P levels and promotes membrane nucleation [20]. During the elongation phase, two ubiquitin-like conjugation systems are involved: the *ATG5-ATG12* pathway and the LC3 (microtubule-associated protein 1A/1B-light chain 3) processing pathway [22]. ATG7 functions as an E1 ubiquitin-activating enzyme, first activating ATG12, which is then transferred to ATG10 to form the ATG5-ATG12 complex. This complex binds to ATG16L, creating an ATG5-ATG12-ATG16L complex that plays a key role in autophagosome membrane elongation and stability. Once an autophagosome is fully formed, the complex dissociates from the membrane [23].

LC3 processing involves the conversion of the microtubule-associated protein light chain 3B (LC3B) into its active form. ATG4 cleaves LC3B to generate LC3B-I, which is activated by ATG3. LC3B-I undergoes conjugation with phosphatidylethanolamine (PE) to form LC3B-II, which is crucial for membrane hemifusion and acts as a mediator in selective degradation pathways. LC3B-II is associated with the phagophore membrane, along with WD repeat domain phosphoinositide-interacting protein 2 (WIPI2), gradually supporting autophagosome formation [24,25]. Once the autophagosome matures, it fuses with lysosomal compartments to form an autolysosome, where the encapsulated substrates are degraded and recycled, thereby completing the autophagy process [26].

## 3. Molecular Targets Regulating Autophagy

### 3.1. FoxO

FoxO proteins play a critical role in regulating autophagy across various tissues under both normal and pathological conditions. They modulate autophagy through several mechanisms, including acting as transcription factors that activate autophagy genes by directly binding to their promoters, interacting with autophagy-related proteins in the cytoplasm, and influencing autophagy via epigenetic regulation [27,28,29]. Key autophagy genes regulated by *FoxO* include Ulk, involved in autophagy initiation; *Becn1* and *ATG14*, associated with nucleation; and Tfeb and Rab7, crucial for autophagosome–lysosome fusion [30,31]. In the context of tissue regeneration, this mechanism is particularly important, as autophagy maintains cellular homeostasis and functional repair while promoting cell proliferation and differentiation.

Stem cells serve as the foundation of tissue regeneration, and the self-renewal and differentiation of stem cells are crucial for effective tissue regeneration. The nuclear localization of FoxO1 and FoxO3 is essential for sustaining autophagic activity and plays a critical role in tissue regeneration. Notably, FoxO1 plays a vital role in sustaining high autophagic activity in embryonic stem cells (ESCs) and adipocytes, promoting both self-renewal and differentiation [32]. The nuclear localization and activity of FoxO1 are regulated by acetylation levels. Under stress conditions, such as starvation, the deacetylases *Sirt1* and *Sirt2* dissociate from FoxO1, leading to increased acetylation, nuclear export, and enhanced interaction with the autophagy-related protein ATG7 [33]. In the liver, a deficiency in Sirt1 leads to increased FoxO3 acetylation, which suppresses the expression of several autophagy genes, including *ATG101*, nucleation-related *ATG14*, and elongation-related genes *Lc3b* and *ATG3* [34]. Thus, both nuclear translocation and exclusion of FoxO are involved in regulating autophagy gene expression, with varying implications for different diseases.

FOXO activity also plays a significant role under certain pathological conditions, particularly through interactions with the PI3K/AKT/mTOR signaling pathway. Starvation further modulates FOXOs by upregulating specific miRNAs, such as *miR-378*, which are associated with the phenotypes of Duchenne and Becker muscular dystrophies. This process enhances FOXO activity by inhibiting Akt kinase, establishing a positive feedback loop that promotes autophagy [35,36].

### 3.2. mTOR

mTORC1 plays a crucial role in regulating autophagy by directly phosphorylating key components of the autophagy machinery, including ULK1, ATG13, and ATG14. This phosphorylation inactivates ULK1 and the PI3K class III complex, which contains ATG14, thereby disrupting the link between nutrient signaling and autophagy [10] (Figure 1). mTORC1 also influences autophagosome expansion via the RACK1-ATG5 signaling pathway. The ATG12-ATG5-ATG16 protein complex facilitates the lipidation of MAP1LC3 (LC3), a crucial step in autophagosome formation, with ATG5 serving as a critical regulatory node controlled by mTORC1 through its interaction with RACK1(Receptor for Activated C Kinase 1) [37].

Furthermore, mTORC1 phosphorylates transcription factors involved in lysosomal biogenesis, including TFEB, TFE3, and MITF. This phosphorylation is mediated by active RAG GTPase heterodimers, which retain these transcription factors in the cytoplasm and prevent the activation of autophagy-related genes [37]. However, mTORC1 activity is suppressed during amino acid starvation, leading to autophagy activation. Under these conditions, TFEB, TFE3, and MITF translocate to the nucleus, where they enhance the transcription of genes involved in autophagy and lysosomal function [38]. This regulatory mechanism highlights the critical role of mTORC1 in balancing autophagy based on nutrient availability. When nutrients are abundant, mTORC1 suppresses autophagy by disrupting the PI3K class III autophagy complex and influencing lysosome formation (Figure 1). This inhibition ensures that resources are directed toward growth and proliferation rather than cellular recycling. However, under nutrient-deficient conditions, mTORC1 activity decreases, thereby supporting autophagy in clearing damaged components. Studies have shown that mTORC1 activity increases during tissue repair, such as in the intestine and liver, promoting cell proliferation and accelerating healing [39,40]. However, in the early stages of tissue regeneration, mTORC1 activity is suppressed [41]. This temporary suppression activates autophagy, which allows for autophagic clearance, creating an optimal environment for new cell growth.

Throughout the regeneration process, autophagy balances resource allocation between cellular recycling and the generation of new cells, ensuring efficient repair and growth. However, overactivity of mTOR may lead to pathological hyperplasia. Therefore, understanding how mTOR and autophagy are coordinated is crucial for developing targeted therapies in regenerative medicine.

### 3.3. AMPK

AMPK plays a critical role in energy homeostasis by regulating the AMP/ATP ratio and autophagy. Mitochondria maintain their function and morphology through fission, fusion, and mitophagy, which removes damaged or unnecessary mitochondria and preserves cellular health. The PINK1/PRKN pathway is a key regulator of mitophagy, and increased PINK1 levels and autophagosome translocation have been observed in the mitochondria of frostbite-damaged muscle [42]. AMPK activates mitophagy by phosphorylating ULK1 at Ser555, facilitating its translocation to the mitochondria [43]. AMPK also phosphorylates ULK1, which in turn activates Beclin1 through phosphorylation, enhancing VPS34 activity [44]. Additionally, phosphorylation of Beclin1 at Ser388 by AMPK disrupts its interaction with Bcl-2, lifting Bcl-2’s inhibition of autophagy [45]. During nutrient deprivation, AMPK inhibits mTORC1 by phosphorylating TSC2 and mTOR, while ULK1 further suppresses mTORC1 via the phosphorylation of Raptor [46,47]. Furthermore, AMPK phosphorylates FOXO, enhancing its nuclear translocation and promoting the expression of autophagy-related genes [48].

## 4. Autophagy and Tissue Repair

As an essential mechanism for preserving cellular health and survival, autophagy plays a vital role in eliminating damaged components and orchestrating tissue regeneration and repair. This function is critical for maintaining homeostasis and combating age-related diseases [17,49]. Autophagy is activated by diverse microenvironmental stressors, such as nutrient deprivation, oxidative stress, and viral infections, enabling cellular adaptation and survival under adverse conditions [50]. Through these mechanisms, autophagy exhibits significant self-repair capabilities in tissues affected by injury or disease. It aids in the survival and differentiation of stem cells, regulates inflammatory responses, and maintains a healthy immune environment [17,20,49,51].

Autophagy plays a vital role in tissue repair across various cellular processes. In neurons, it supports axonal regeneration by stabilizing microtubules, which are essential for restoring function after injury. In the intestinal system, autophagy ensures the proliferation of stem cells. Studies in fruit flies have shown that autophagy deficiency leads to the nuclear accumulation of Chk2, resulting in increased DNA damage and cell cycle arrest [52]. Additionally, autophagy helps maintain the health of hematopoietic stem cells (HSCs) by protecting them from metabolic stress, thereby contributing to long-term tissue regeneration and longevity [49,53]. Autophagy is also crucial for zebrafish tail fin regeneration, aiding in both the dedifferentiation and redifferentiation of mature cells [54]. Given its significant role in tissue regeneration, autophagy represents a promising target for the repair of damaged or dysfunctional tissues.

## 5. Autophagy Promotes Muscle Regeneration

### 5.1. The Process of Muscle Regeneration

Muscle regeneration is a complex process that involves several key stages: necrosis of damaged muscle fibers, activation of satellite cells, proliferation of these activated stem cells, differentiation of myoblasts, and the formation and remodeling of new muscle fibers (Figure 2). Lineage tracing studies have identified a subset of PAX+ satellite cells that express *PAX7* (Paired Box Gene 7) in both quiescent and activated states [55]; however, quiescent cells specifically lack MyoD (myogenic differentiation factor) expression (Figure 2). Upon activation, satellite cells co-expressing PAX7 and MyoD—early indicators of myogenic commitment—exit quiescence, differentiate into myocytes, and mature into muscle fibers. Myogenic regulatory factors (MRFs) such as MyoD, Myf5, Myogenin, and MRF4, which operate downstream of PAX3 and PAX7, are crucial for promoting myogenic differentiation and cell cycle exit [56]. During embryonic development, MRFs guide muscle progenitor cells (MPCs) to differentiate into muscle fibers, meeting the developmental needs of the muscles of the head, trunk, and limbs [57]. MyoD, a key myogenic determination protein, is expressed early during myogenesis but can be inhibited by PAX7, which suppresses Myogenin-induced myogenic conversion [58].

Satellite cell activity during skeletal muscle regeneration is tightly regulated by a time-dependent inflammatory response characterized by a shift between pro-inflammatory and anti-inflammatory macrophage phenotypes [59] (Figure 2). Regeneration begins with the activation of the complement system by necrotic fibers and mast cell degranulation, which releases pro-inflammatory cytokines that attract immune cells. Neutrophils and pro-inflammatory macrophages clear cellular debris and necrotic tissue. Pro-inflammatory macrophages secrete cytokines such as TNF-α, IL-6, and IL-1β, sustaining inflammation at the injury site [60] (Figure 2). TNF-α, in particular, stimulates MuSCs to activate and proliferate rapidly without differentiating [61]. Around four days post-injury, the microenvironment shifts to an anti-inflammatory state, coinciding with peak MuSC numbers. Regulatory T cells (Tregs) infiltrate this area and secrete IL-10, which promotes the transition of macrophages from the M1 to M2 phenotype. M2 macrophages then release cytokines, such as IL-4, IL-10, and IL-13, facilitating MuSC differentiation and fusion [60,62] (Figure 2). Finally, the differentiated satellite cells contribute to muscle repair by fusing with existing fibers or forming new myotubes. This well-coordinated inflammatory response is essential for effective skeletal muscle regeneration.

### 5.2. Autophagy Regulates the Function of Satellite Cells

#### 5.2.1. Autophagy Counteracts Apoptosis in Muscle Satellite Cells

The transition of satellite cells from quiescence to activation involves significant metabolic shifts, particularly from oxidative metabolism to glycolysis [63]. For example, miR-378 enhances glycolytic activity, which enriches *Pax7^Hi^* satellite cells and delays their activation [64]. Similarly, reactive oxygen species (ROS) levels influence satellite cell behavior: higher ROS levels promote differentiation, while lower ROS levels help maintain quiescence [65]. Autophagy plays a critical role in supporting these processes by recycling cellular components to generate energy. Additionally, it regulates oxidative stress, maintaining cellular homeostasis, which creates favorable conditions for effective cell regeneration. The regulation of satellite cell function by autophagy is illustrated in Figure 3.

In autophagy-deficient mice, satellite cells are highly susceptible to apoptosis. Specifically, ATG7 knockdown in neonatal satellite cells (nSCs) impairs their proliferation and differentiation, leading to the accumulation of NFE2L2/NRF2(Nuclear factor erythroid 2-related factor 2), which interferes with the DDIT3/CHOP(DNA Damage Inducible Transcript 3)pathway [66]. This disruption suppresses the GH-IGF1 pathway, a key regulator of muscle development. In adult mice, ATG7 deficiency results in elevated levels of reactive oxygen species (ROS) and mitochondrial dysfunction, ultimately reducing the number of satellite cells and accelerating aging [66,67]. These findings emphasize the vital role of autophagy in maintaining muscle health and the regenerative potential of satellite cells in both neonatal development and adult muscle repair.

Autophagy also plays a key role in protecting satellite cells under stressful conditions, such as surgical trauma. For instance, following anterior cruciate ligament (ACL) reconstruction, there is a marked depletion of satellite cells in the quadriceps due to increased apoptosis of **PAX7+** satellite cells (**PAX7+/TUNEL+**) [68]. Recent studies have found that autophagosome flux in the quadriceps muscle increases during ACL injury recovery (14–28 days), accompanying the muscle regeneration process [69]. This increase may be attributed to the decreased preservation of satellite cell populations, a function normally safeguarded by autophagy.

As the body ages, the balance between autophagy and apoptosis becomes increasingly dysregulated, largely due to a progressive decline in autophagic activity, coupled with increased susceptibility to apoptotic signaling. One key regulator in this process is **NIMA-related kinase 2 (Nek2)**, which decreases in aged muscle stem/progenitor cells [70]. Nek2 deficiency contributes to abnormal centrosome formation, triggering apoptotic pathways via the activation of **p53** [70]. The p53 pathway is a central mediator of cell death, and its hyperactivation, when autophagy is insufficient, accelerates tissue damage and contributes to organismal decline [71].

Under many conditions, autophagy plays a protective role against apoptosis by clearing damaged organelles to reduce metabolic stress while providing essential nutrients during starvation [72,73]. This function is particularly important in stem cells, where the energy demands during the transition from quiescence to proliferation render them vulnerable to apoptosis. Autophagy acts as an energy reserve to support stem cell survival during periods of metabolic stress. The **AMPK/p27Kip1** pathway plays a key role in balancing autophagy and apoptosis, particularly in stem cells under metabolic stress [74,75]. **p27Kip1**, a cyclin-dependent kinase inhibitor, regulates cell fate by mediating autophagy or apoptosis based on its subcellular localization and phosphorylation state. Phosphorylation of **p27Kip1** at **Thr198**, mediated by **AMPK**, activates autophagy and prevents apoptosis [76]. Studies have shown that activation of this pathway can enhance the survival and proliferation of aged MuSCs, reduce cellular senescence, and improve transplantation outcomes [75]. Targeting the **AMPK/p27Kip1** pathway offers a promising therapeutic strategy for enhancing muscle regeneration in elderly individuals by restoring the balance between autophagy and apoptosis.

#### 5.2.2. Autophagy Balances Satellite Cell Quiescence and Activation

Under normal conditions, satellite cells remain in a quiescent state, and their cell cycle is temporarily suppressed. This preservation of the stem cell pool prevents unnecessary energy expenditure in the absence of injury. Upon muscle injury, signals from the local environment activate satellite cells [1,2]. Autophagy plays a critical role in balancing the quiescence and activation of these cells.

When autophagy is impaired, satellite cell quiescence is disrupted, leading to either premature or delayed activation of MuSCs, which can contribute to failed muscle regeneration. For instance, denervation-induced muscle atrophy is often accompanied by the premature activation of MuSCs, along with elevated expression of early activation markers such as PAX7 and MyoD [77]. Studies have shown that following denervation, there is an early increase and later decline in autophagic flux in mouse muscle, accompanied by mitochondrial dysfunction [78]. It remains unclear whether these changes are directly related to impairments in the function of muscle satellite cells (MuSCs). However, premature activation often leads to apoptosis, hindering effective muscle repair. In contrast, autophagy-deficient mice treated with cardiotoxin (CTX) show impaired stem cell activation, characterized by a higher proportion of stem cells (PAX7+/MyoD−) in regenerating muscles and a lower proportion of MPC(PAX7+/MyoD+) or committed myogenic cells (PAX7−/MyoD+) [79].

With aging, autophagy plays a crucial role in maintaining stem cell quiescence. Stem cells that fail to return to a quiescent state for extended periods tend to differentiate rather than maintain self-renewal. Loss of quiescence leads to a decline in both the number and function of adult stem cells [80]. A high basal autophagic flux is critical for satellite cells to maintain their quiescent state by continuously clearing intracellular toxic substances and preserving cellular homeostasis [67]. In addition, autophagy defects lead to the accumulation of reactive oxygen species (ROS) in damaged mitochondria [81]. During aging, this is further exacerbated by a systemic decline, reduced mitochondrial energy production, and mitochondrial DNA damage, which deprive satellite cells of the energy required for activation and differentiation [8,67,75].

Autophagy may also influence satellite cell activation by regulating nitric oxide (NO), a crucial messenger in skeletal muscle [82]. NO facilitates the removal of damaged muscle cells by immune cells while stimulating the release of hepatocyte growth factor (HGF) and other growth factors [83]. It supports satellite cell activation by enhancing the release of growth factors and protecting against oxidative stress [82,84]. In endothelial cells, autophagy—particularly mitophagy, which degrades dysfunctional mitochondria—enhances the efficiency of NO synthesis [85,86]. Moreover, autophagy helps regulate NO bioavailability by reducing oxidative stress [86,87], which may further support satellite cell activation.

Thus, autophagy is essential not only for cellular degradation but also for preserving the regenerative capacity of muscle stem cells throughout life. Understanding how autophagy balances quiescence and activation opens potential therapeutic avenues to combat muscle degeneration.

### 5.3. Autophagy Promotes the Differentiation of Myoblasts

The final differentiation of skeletal muscle is characterized by the fusion of myoblasts into multinucleated myotubes. Muscle cells cease active division and arrest in the G0 phase of the cell cycle. Differentiated muscle cells that successfully exit the cell cycle acquire an anti-apoptotic phenotype and become mature myotubes [88,89].

#### 5.3.1. Autophagy/Mitophagy Regulates Mitochondrial Remodeling

Satellite cells undergo significant metabolic changes during quiescence, proliferation, and differentiation. As myogenic cells differentiate into muscle fibers, their metabolism shifts to oxidative phosphorylation to meet the high energy demands of mature muscles [90,91].

During the early stages of myogenic differentiation, mitophagy plays a crucial role in remodeling the mitochondrial network to support the bioenergetic requirements of differentiation [92,93,94]. Autophagy is upregulated during this phase, characterized by increased Bnip3 expression and reduced levels of peroxisome proliferator-activated receptor gamma coactivator 1-alpha (PPARGC1A/PGC1α) and mitochondrial proteins [95]. This process helps clear damaged or unnecessary mitochondria, optimizing the mitochondrial network for the cell’s evolving metabolic requirements.

Another study confirmed that upregulation of autophagy is closely linked to mitophagy, facilitating the clearance of existing mitochondria while stimulating mitochondrial biogenesis [94]. The reorganization of the mitochondrial network involves the fusion protein OPA1, which helps reform mitochondrial structures after fragmentation driven by DNM1L/DRP1 (Dynamin-1-like protein) and subsequent clearance via SQSTM1 [90]. This dynamic mitochondrial remodeling is crucial for the proper differentiation of satellite cells into functional muscle fibers, highlighting the importance of coordinated autophagy and mitophagy in muscle development and regeneration.

#### 5.3.2. The Role of Apoptosis and Autophagy in Myogenic Differentiation

Apoptosis is essential for skeletal muscle differentiation. Zinc finger protein 422 (Zfp422) disrupts these processes by regulating apoptosis through EphA7 modulation [96] (Nie et al., 2020). Although EphA7 promotes both autophagy and apoptosis [97], its specific role in muscle autophagy remains unclear. Several apoptosis-related proteins are involved in the differentiation of myogenic cells. Caspase 3, a key enzyme in the apoptotic pathway, promotes myogenic cell fusion and myotube formation by interacting with mammalian sterile 20-like kinase (MST1) [98]. Caspase 9, activated by the apoptosome complex, plays a crucial role in the mitochondrial death pathway. Inhibition of caspase 9 using shRNA can prevent caspase 3 activation, thereby hindering myogenic fusion [99]. Overexpression of Bcl-xL, an anti-apoptotic protein, suppresses caspase 9 activation, underscoring the complex interplay between apoptotic regulation and muscle differentiation [99]. However, while caspase activation is necessary for differentiation, excessive activation can lead to cell death [100,101]. Thus, the regulation of apoptotic signaling during differentiation must be tightly controlled to ensure a balance between promoting differentiation and preventing apoptosis. The precise mechanisms regulating apoptotic signaling during skeletal muscle differentiation remain unclear.

Research indicates that defects in autophagy during myogenic differentiation can increase the likelihood of apoptosis [102,103]. Myogenic cells with stable *ATG7* knockdown exhibit compromised mitochondrial membrane integrity, increased levels of the DNA fragmentation marker p-H2AFX (phosphorylated H2A histone family member X), and elevated caspase 3 (CASP3) activity [102,104]. Loss of mitochondrial membrane integrity triggers apoptotic signaling, with key factors including cytochrome c (CYCS), which is released into the cytosol and activates the apoptosome, leading to caspase-dependent apoptosis. In the early stages of differentiation, CASP9 activity is higher in Bnip3 knockout myogenic cells, indicating enhancement of the mitochondria-mediated apoptotic pathway [95]. These disruptions underscore the critical role of autophagy in maintaining mitochondrial health and regulating apoptotic signaling during muscle differentiation, highlighting the need for balanced autophagic activity to support proper myogenic progression.

## 6. New Directions in Autophagy Regulation of Muscle Regeneration

### 6.1. Muscle Regeneration and Macrophages

Muscle damage from trauma, burns, toxins, and other factors triggers an inflammatory response. Resident macrophages release neutrophil chemoattractants CXCL1 and CCL2, promoting neutrophil infiltration. Genetic ablation of *Ccl2* or its receptor *Ccr2* in mice reduces the number of macrophages in injured muscles, leading to fewer newly formed muscle fibers and highlighting the critical role of macrophages in muscle repair and regeneration [105,106]. Blood monocytes infiltrate damaged tissue and initially differentiate into Ly6C^high^ macrophages, which phagocytose debris and express pro-inflammatory factors. These macrophages then shift to an anti-inflammatory Ly6C^low^ phenotype. Based on their activation state, macrophages are classified as M1 (pro-inflammatory) or M2 (anti-inflammatory), similar to T lymphocyte categorization [60,107].

Chazaud et al. demonstrated the dynamic transformation and dual role of macrophages during muscle injury repair. Specifically, M1 macrophages are involved in clearing necrotic tissue, while M2 macrophages promote the proliferation and differentiation of myogenic cells [59,108,109,110,111]. Additionally, macrophages mediate the resolution of inflammation and facilitate muscle regeneration through continuous phenotypic changes [111]. Ly6C^high^ macrophages are associated with a pro-inflammatory state that supports muscle regeneration and limits the expansion of fibroadipogenic progenitors (FAPs) involved in fibrosis. Conversely, Ly6C^low^ macrophages exhibit an anti-inflammatory phenotype, promoting MPC differentiation, extracellular matrix remodeling, and angiogenesis, all of which are key processes in effective muscle repair [61,112,113]. Regulation of the shift from a pro-inflammatory to a pro-repair macrophage state offers a therapeutic strategy for enhancing muscle regeneration. For example, the AMPK signaling pathway plays a crucial role in the polarization of macrophages [114]. Annexin A1 (ANXA1) in macrophages induces a pro-repair phenotype, reduces inflammation, and restores muscle homeostasis through activation of AMPK [114,115]. Additionally, lactate supplementation in pfkfb3-deficient endothelial cells can restore M2-like macrophage polarization via an MCT1-dependent mechanism, aiding ischemic muscle regeneration [113]. These findings highlight the potential for modulating macrophage phenotypes to improve muscle repair and address muscle degeneration under various conditions.

### 6.2. Autophagy Regulates Macrophage Polarization

Autophagy is crucial for the differentiation of monocytes into macrophages and for acquiring their phagocytic functions, likely involving the ULK1- and ATG7-dependent pathways. Studies have shown that autophagy suppresses M1 polarization of macrophages and plays a significant role in the treatment of chronic inflammation and fibrosis [116]. Recent evidence indicates that autophagy regulates macrophage M1/M2 polarization under various inflammatory conditions. Blocking autophagy, particularly through ATG7 knockout, increases reactive oxygen species (ROS) in macrophages, promoting an M1 inflammatory phenotype [117]. Additionally, ATG5 knockdown abolishes the inhibitory effect of spermidine (SPM) on M1 polarization, thereby promoting M2 polarization [118].

Polyunsaturated fatty acids, such as docosahexaenoic acid (DHA), enhance M2 macrophage marker expression via the p38 MAPK signaling pathway and autophagy [119]. Targeting macrophage phenotype switching is a key focus in muscle regeneration therapy, although direct links between autophagy regulation of macrophage polarization and muscle regeneration have yet to be established. We propose that modulating autophagy to alter macrophage inflammatory phenotypes could significantly enhance muscle regeneration.

## 7. Advances in Research on Autophagy in Regenerative Deficiency Diseases

### 7.1. Sarcopenia

In sarcopenia, satellite cells decline in number and function owing to intrinsic and extrinsic factors. Extrinsic factors include the muscle microenvironment and external signals, whereas intrinsic factors include oxidative stress, DNA damage, and autophagy [82]. In aging quiescent stem cells, increased ROS levels cause DNA damage and accelerate the aging process [120]. The downregulation of *p16(INK4a)*, also known as Cdkn2a, is linked to the irreversible loss of quiescence in satellite cells. This downregulation is mediated by the *PRC1* (Polycomb Repressive Complex 1), which ubiquitinates lysine 119 on histone H2A, thereby suppressing p16(INK4a) activity [8,121]. ROS also contribute to the irreversible loss of quiescence and promote senescence in satellite cells through the epigenetic regulation of *INK4a* [8].

SIRT proteins partially fulfill energy requirements by activating the FoxO3a pathway [122,123]. *Deaf1*, when overexpressed, deforms the Epidermal Autoregulation Factor-1, a key regulator of muscle regeneration, and inhibits autophagy. This inhibition leads to protein accumulation and apoptosis of muscle stem cells (MuSCs). Deaf1 is transcriptionally repressed by FoxO proteins, which activate autophagy-related genes such as ATG16I1 and Pik3c3 [124]. Additionally, knockout of ATG7 impairs neonatal satellite cell regeneration via the GH-IGF-1(Growth Hormone Insulin-like Growth Factor 1) axis. IGF-1 plays a crucial role in satellite cell proliferation and differentiation by enhancing mitophagy and mitochondrial function, largely through the upregulation of PGC1-α expression [66].

### 7.2. Duchenne Muscular Dystrophy (DMD)

Duchenne Muscular Dystrophy (DMD) is caused by mutations in the dystrophin gene, leading to increased muscle fiber fragility, loss of satellite cell polarity, and functional impairment, which compromise muscle regeneration [125]. In the early stages of DMD, satellite cell-mediated regeneration can temporarily counteract muscle fiber degeneration; however, as the disease progresses, muscle tissue increasingly undergoes fibrosis and fat infiltration [125,126]. This progression correlates with initial autophagy activation, which later becomes impaired [9].

Insufficient mitophagy is critical in the pathogenesis of DMD. In dystrophic animal models, dystrophin mutations are linked to reduced mitochondrial respiratory capacity, enlarged mitochondria, and decreased membrane potential in muscle cells [127,128,129]. Dysfunctional mitochondria in DMD-affected muscles are not efficiently removed by autophagy, affecting both muscle fibers and muscle stem cells (MuSCs) [130]. During regeneration, energetically compromised mitochondria are transferred to muscle fibers when differentiated satellite cells fuse with them, worsening mitochondrial dysfunction in the tissue [131].

FoxO transcription factors are vital in regulating autophagy by positively influencing the expression of genes related to autophagy induction (ULK1 and ULK2) and autophagosome–lysosome fusion (TFEB) [27,30]. The transcriptional activity of TFEB is inhibited by mTORC1 [132]. In mdx mice, the skeletal muscle shows downregulation of autophagy-related genes, enhanced Akt-mTORC1 signaling, and reduced nuclear levels of FoxO1 and FoxO3a, likely due to decreased FoxO and TFEB activity from phosphorylation [133]. Additionally, Src kinase links Nox2-specific oxidative stress with autophagy defects, impairing lysosome formation and starvation-induced autophagy via the PI3K/Akt/mTOR pathway [134]. A recently discovered regulator, Islr, is essential for the classical Wnt signaling pathway, which is crucial for satellite cell differentiation. Disheveled-2 (Dvl2), a key Wnt signaling component, interacts with Islr to promote muscle regeneration. Islr protects Dvl2 from autophagy-mediated degradation, thereby activating Wnt signaling [135]. These insights into the molecular mechanisms offer potential targets for the development of new therapies for DMD.

### 7.3. Myotonic Dystrophy Type 1 (DM1)

Myotonic Dystrophy Type 1 (DM1) is a genetic disorder caused by an expanded CTG repeat in the DMPK gene, leading to nuclear RNA foci and reduced MBNL1 levels [136,137]. Studies have shown that autophagy is elevated in satellite cells (SSCs) and arterial primary SSCs of DM1 patients, indicated by increased LC3II/LC3I ratios and decreased levels of P62 [138]. Overexpression of MBNL1 increases phosphorylated mTOR, promoting SSC proliferation by inhibiting autophagy. This effect can be reversed by rapamycin, an mTOR inhibitor that counteracts MBNL1’s actions [19] (Song, 2020).

The RNA-binding protein Musashi-2 (MSI2) exacerbates DM1 progression by inhibiting MicroRNA-7 (miR-7), thereby reducing miR-7’s suppression autophagy [139]. Recent studies have identified *MYTHO*, a FoxO-dependent gene with reduced expression in DM1 patients, as a potential therapeutic target. Prolonged knockdown of MYTHO impairs autophagy and leads to muscle weakness, potentially due to activation of the mTORC1 signaling pathway [140]. MYTHO may interact with WIPI2, a protein involved in autophagosome formation, and could play a role in phagophore assembly. However, the precise relationship between MYTHO and autophagy requires further study [141,142]. These findings highlight the complex regulatory landscape of autophagy in DM1 patients and underscore the need for ongoing research into these molecular pathways to identify novel therapeutic approaches.

## 8. Therapeutic Directions

Several studies have shown that insufficient autophagy impairs muscle regeneration, but excessive autophagy can also lead to cell death [143]. Research has indicated that autophagic flux in stem cells gradually increases during differentiation; however, controlling the timing and extent of autophagy activation remains a significant challenge. Various compounds and biological agents that target autophagy, mainly inducers such as spermidine, rapamycin, pomegranate polyphenols, and metformin, have been developed and are promising for treating muscle diseases associated with autophagy deficiency [144,145,146]. For example, spermidine, a drug gaining attention in clinical trials, offers mitochondrial protection and anti-inflammatory effects, with a unique role in preventing stem cell aging. However, studies show that its interaction with mTOR signaling varies between young and aged adipose tissues [147]. Additionally, its impact on neurotransmitter release under fasting or caloric restriction remains unclear, limiting its clinical development [148]. Similarly, pomegranate polyphenols regulate autophagy via antioxidant mechanisms, but their interaction with neurotransmitters like dopamine raises concerns about potential cytotoxicity, necessitating careful evaluation of safe dosages for therapeutic use [149,150]. Despite the potential of these therapeutic agents, aligning the effects and timing of autophagy inducers with the dynamic needs of muscle regeneration is complex due to the multifaceted role of autophagy in this process. Caution is warranted regarding the risks of excessive or prolonged autophagy activation, which could negatively affect muscle regeneration.

Cell therapy represents a promising approach and causes less damage to healthy tissues with fewer side effects compared to traditional drug treatments. Various cell types including MuSCs, bone marrow-derived mesenchymal stem cells (BM-MSCs), adipose-derived stem cells (ADSCs), and induced pluripotent stem cells (iPSCs) are utilized to enhance skeletal muscle regeneration [151]. Several of these cell types have shown potential in muscle repair through the regulation of autophagy (Table 1). Studies have shown that stem cell transplantation in mdx mice promotes muscle regeneration and improves muscle function; however, the effects of transplanted stem cells on autophagy in DMD muscles remain unexplored. Additionally, mesenchymal stem cells (MSCs), including tonsil-derived MSCs (TMSCs), have been shown to differentiate into skeletal muscle cells (TMSC–myocytes), highlighting their potential as therapeutic resources for DMD [152]. However, a previous study indicated that co-culturing iPSCs with C2C12 myoblasts can inhibit autophagy and prevent the atrophy of C2C12 myotubes under oxidative stress [153]. These findings suggest that stem cell transplantation promotes muscle regeneration and functional recovery, potentially through the regulation of autophagy.

Biomaterials have become increasingly prominent in tissue regeneration owing to their excellent biocompatibility and multifunctionality. For instance, decellularized extracellular matrix (ECM) scaffolds co-seeded with adipose-derived stem cells and L6 cells have shown effectiveness in treating large muscle defects and promoting muscle regeneration [157] (Liang, 2024). Compared to traditional drugs and genetic manipulation, biomaterials offer superior advantages in regulating autophagy, enabling tissue regeneration in a time- and space-dependent manner while enhancing drug delivery with improved targeting. For example, self-assembling nanostructures can co-assemble spermidine and mTOR siRNA, enhancing autophagy in macrophages [158]. Similarly, silver nanoparticles (AgNPs) loaded onto TiO_2_ nanotubes activate autophagy and promote bone tissue repair by inducing M2 macrophage polarization through the release of ultra-low doses of silver ions [159]. Additionally, nanoparticles can enhance the overexpression of TFEB, a key transcription factor for lysosomal function, thereby boosting the expression of ATG and lysosomal genes [160]. In muscle tissue repair, nanomaterials offer unique advantages for targeted autophagy regulation, facilitating cell- and spatial-specific targeting that could improve therapeutic outcomes in muscle regeneration.

## 9. Outlook and Conclusions

This review focuses on the molecular mechanisms of autophagy in muscle regeneration. Since recent studies on muscle regeneration have predominantly centered around chronic muscle diseases, most of the examples cited in this paper are drawn from research on conditions such as sarcopenia and muscular atrophy. This focus somewhat overlooks the relationship between autophagy and acute muscle injuries.

Chronic injuries are typically associated with sustained metabolic or oxidative stress, leading to prolonged or impaired autophagy [161,162]. In contrast, acute injuries (e.g., intense exercise, contusions, or tears) occur under sudden stress, triggering a brief but robust activation of autophagy to rapidly clear damaged components [79,163]. While autophagy serves a compensatory and beneficial role in acute injuries, its role in chronic muscle diseases is more complex. On the one hand, autophagy alleviates oxidative stress and mitigates aging; on the other hand, dysregulated autophagy may contribute to muscle fibrosis and loss of muscle mass [164,165]. The following table (Table 2) summarizes the differences in autophagy’s roles in acute versus chronic muscle injuries. Understanding these distinctions is essential for designing therapeutic interventions to optimize muscle repair and prevent chronic muscle wasting.

In summary, autophagy-deficient models are invaluable for understanding the role of autophagy in muscle regeneration. However, the function of autophagy varies across different stages of satellite cell activity—including quiescence, activation, differentiation, and myogenesis—and the precise molecular mechanisms remain unclear. Further research is essential to elucidate the intensity and impact of autophagy on satellite cell physiology and its interaction with other physiological processes. The development of cell therapies and biomaterials presents promising avenues for replicating or enhancing normal muscle regeneration, offering new hope for the treatment of complex muscle regeneration disorders.

## Figures and Tables

**Figure 1 ijms-25-11901-f001:**
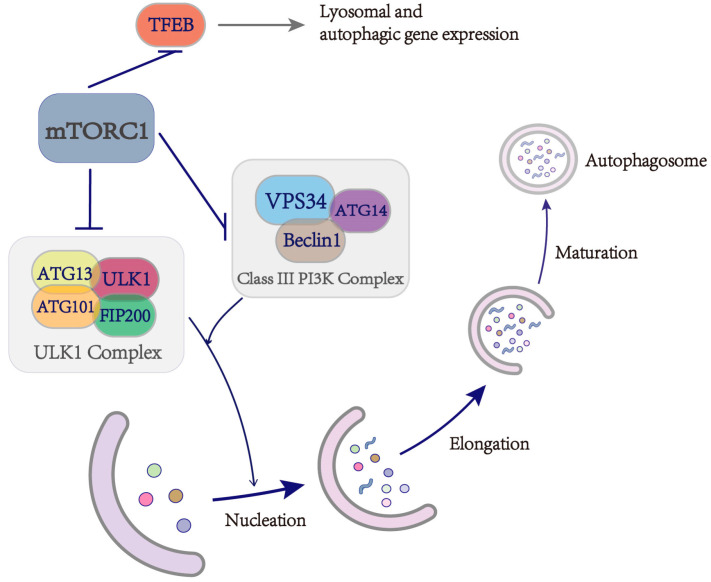
Regulation of the autophagy process by mTORC1. The graph illustrates how mTORC1 regulates the autophagy process by inhibiting key components involved in autophagosome formation. Specifically, mTORC1 suppresses the activation of the ULK1 complex and the PI3K complex, both of which are essential for initiating autophagy. Additionally, mTORC1 downregulates TFEB activity, thereby reducing the transcription of autophagy-related genes.

**Figure 2 ijms-25-11901-f002:**
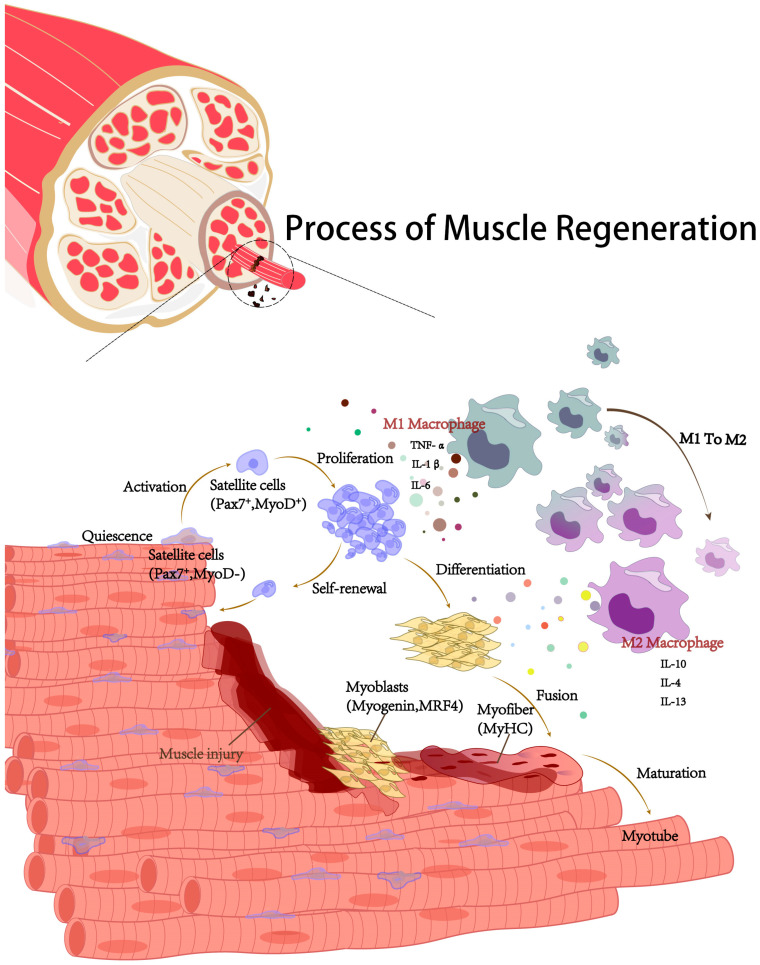
Process of muscle regeneration. During the initial phase of inflammation of muscle regeneration, Pax7+ satellite cells are activated and proliferate, expressing MyoD. A subset of these cells downregulate MyoD and return to a quiescent state, while others exit the cell cycle and continue to differentiate into myoblasts (expressing Myogenin, MRF4), then fuse to form multinucleated myotubes (expressing MyHC) and further develop into new myofibers. These events coincide with the inflammatory shift of macrophages, where muscle cell regeneration is accompanied by a transition of macrophages from a pro-inflammatory phenotype M1 (secreting cytokines: TNF-α, IL-1β, IL-6, promoting satellite cell activation and proliferation) to an anti-inflammatory phenotype M2 (secreting cytokines: IL-10, IL-4, IL-13, promoting satellite cell differentiation).

**Figure 3 ijms-25-11901-f003:**
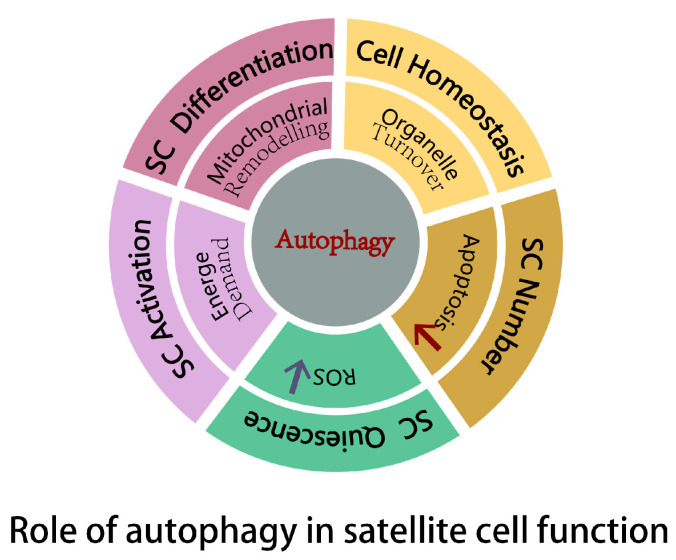
Role of autophagy in satellite cell function. Autophagy maintains cellular homeostasis by clearing damaged organelles. Autophagy preserves the number of satellite cells by antagonizing apoptosis. Autophagy sustains the quiescence of satellite cells by reducing reactive oxygen species (ROS). Autophagy promotes the activation of satellite cells by supplying energy. Autophagy facilitates the differentiation of satellite cells by mitochondrial remodeling.

**Table 1 ijms-25-11901-t001:** Summary of autophagy and stem cell therapy in muscle diseases.

Stem Cell Type	Autophagy Induction	Model	Mechanism of Autophagy Involvement	Therapeutic Effects	Key Points	Reference
Human umbilical MSCs (hucMSCs)	Increased	Diabetic db/db mice, high-fat diet (HFD) fed mice	Enhances AMPK/ULK1-mediated autophagy	Improves muscle atrophy in diabetes and obesity		[154]
Adipose-derived mesenchymal stem cells (AD-MSCs)	Increased	Peripheral arterial disease (PAD) mice	Hypoxia-induced AMPK-mTOR-ULK1 and reoxygenation-related Akt-Bcl2-Beclin1 signaling induce autophagy	Restores limb function and blood perfusion in PAD mice	Protective autophagy enhances AD-MSC survival and inhibits cell death	[155]
Clinical-grade human umbilical cord-derived mesenchymal stem cells (hUC-MSCs)	Increased	SAMP8 mice and D-galactose-induced aging mice	Increases autophagy via p16-Rb/p53-p21 pathway	Improves muscle strength and restores skeletal muscle morphology in aging mice	hUC-MSCs promote extracellular matrix expression, satellite cell activation, and autophagy	[156]
Tonsil-derived MSCs (TMSCs)	Increased	mdx mouse (DMD model)		Enhances muscle repair in Duchenne Muscular Dystrophy (DMD)	TMSCs induce autophagy to support skeletal muscle regeneration in mdx mice	[152]
Induced pluripotent stem cells (iPSCs)	Decreased	C2C12 myoblasts exposed to hypoxia	Suppresses autophagy via AMPK/mTOR/ULK1 pathway	Improves myotube formation under hypoxic conditions	Suppression of autophagy promotes cell survival and prevents myotube degradation	[153]

**Table 2 ijms-25-11901-t002:** Comparison of autophagy roles in acute vs. chronic muscle injuries.

Aspect	Acute Muscle Injury	Chronic Muscle Injury
Activation	Rapid, transient	Sustained, chronic
Trigger	Exercise [166], trauma [167], ischemia [168]	Dystrophy [9,169], cachexia [170], aging [16,171]
Role	Protective, aids recovery	Mixed (beneficial and pathological)
Regulatory Pathways	AMPK activation, mTOR inhibition	Inflammation, oxidative stress
Impact on Muscle	Promotes repair and regeneration	Contributes to atrophy and fibrosis [16]

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
