# Peer review of "Autophagy in Muscle Regeneration: Mechanisms, Targets, and Therapeutic Perspective"

_ijms, 2024, doi:10.3390/ijms252211901_

Round 1
Reviewer 1 Report
Comments and Suggestions for Authors
The review manuscript by Chu et al., titled "Autophagy in Muscle Regeneration: Mechanisms, Targets, and Therapeutic Perspective," provides a comprehensive overview of the role of autophagy in muscle regeneration, particularly in regulating satellite cell quiescence, activation, and differentiation. The authors also discuss the involvement of autophagy in macrophage polarization and its broader impact on muscle physiology. Toward the end, they explore therapeutic strategies aimed at modulating autophagy to enhance muscle regeneration.
This manuscript is well-written, presenting its key points clearly and emphasizing the importance of understanding autophagy to improve muscle regeneration. However, I have several suggestions that may further improve the quality of the manuscript:
- The authors do not address how autophagy affects other non-muscle cells critical for muscle homeostasis, such as fibroadipogenic progenitors and endothelial cells.
- It would be beneficial for the authors to clearly distinguish between the roles of autophagy in acute versus chronic muscle injuries. Currently, they shift between these processes without offering a clear, structured explanation for the reader.
- The section on the therapeutic modulation of autophagy is the weakest part of the review. I recommend rewriting this portion with a stronger focus on potential pharmacological strategies for targeting autophagy.
- Lastly, the manuscript uses inconsistent terminology and abbreviations for myogenic progenitor cells. I suggest standardizing the nomenclature throughout to improve clarity.
Author Response
ear Reviewer,
Thank you for your review and valuable suggestions on our paper. Based on your feedback, we have revised the manuscript, and the specific changes are marked in blue for your convenience.
Comments 1:The authors do not address how autophagy affects other non-muscle cells critical for muscle homeostasis, such as fibroadipogenic progenitors and endothelial cells.
Response 1:Thank you for your advice. However, the primary focus of our review is to delve into the role of autophagy specifically within muscle cells. The topic of autophagy in other non-muscle cells has been thoroughly discussed in other reviews, such as (Chen et al., 2022).To avoid redundancy and maintain the clarity and depth of our analysis, we have chosen to limit the scope of this review to the direct role of autophagy in muscle cells.
Bibliography:
Chen, W.; Chen, Y.; Liu, Y.; Wang, X. Autophagy in muscle regeneration: potential therapies for myopathies. J Cachexia Sarcopenia Muscle 2022, 13(3), 1673-1685.
Comments 2:It would be beneficial for the authors to clearly distinguish between the roles of autophagy in acute versus chronic muscle injuries. Currently, they shift between these processes without offering a clear, structured explanation for the reader.
Response 2:Thank you very much for your insightful feedback. We appreciate your suggestion to distinguish between the roles of autophagy in acute and chronic muscle injuries, as this is essential for designing therapeutic interventions to optimize muscle repair or prevent chronic muscle wasting.In response to your comment, we have revised the discussion section by adding a more structured explanation. Specifically, we now provide a table that focus on the role of autophagy in acute injuries (e.g., trauma, exercise-induced damage) and chronic conditions (e.g., muscular dystrophies, sarcopenia). In these sections, we discuss the differences in autophagy activation, functional roles, regulatory pathways, and their impact on muscle health.(Page 22-23,line 481-496)
Comments 3:The section on the therapeutic modulation of autophagy is the weakest part of the review. I recommend rewriting this portion with a stronger focus on potential pharmacological strategies for targeting autophagy.
Response 3:Thank you very much for your valuable feedback. We appreciate your suggestion to strengthen the section on the pharmacological modulation of autophagy. In response to your suggestion, we have removed the section on treatment prospects from the discussion and rewritten it as a separate paragraphto enhance the clarity and flow of the manuscript.We now include additional examples of pharmacological interventions and compare them with cell-based therapies.However,while we recognize the importance of pharmacological strategies, the primary focus of this part is to explore the emerging potential of cell therapies and biomaterials for modulating autophagy, as we believe these approaches offer innovative solutions beyond traditional drug treatments.(Page 19,line 436-456)
Comments 4:Lastly, the manuscript uses inconsistent terminology and abbreviations for myogenic progenitor cells. I suggest standardizing the nomenclature throughout to improve clarity.
Response 4:Thank you for your careful review and valuable feedback. We appreciate your suggestion regarding the inconsistent terminology and abbreviations for myogenic progenitor cells. In response, we have thoroughly reviewed the manuscript and standardized the nomenclature throughout to ensure consistency and improve clarity.
Thank you again for your insightful comments, which have contributed to improving the quality of our paper.
Reviewer 2 Report
Comments and Suggestions for Authors
This review by Yun Chu and colleagues summarizes recent progress in the field of autophagy, detailing its role in cellular processes and muscle regeneration and exploring new therapeutic avenues for the future treatment of muscle disorders. The reviewer acknowledges the effort put into this comprehensive review and provides the following comments highlighting concerns and areas for improvement.
1. The review is structured in a way that lacks cohesion between paragraphs, resulting in a descriptive list rather than an integrated analysis of the various aspects of autophagy biology. While the sections are clearly titled, some transitions between topics, such as between general autophagy mechanisms and specific diseases, feel abrupt. A better flow or linking between sections could improve the reading experience and the authors should create more critical and speculative connections between these aspects.
2. Certain parts of the review are underdeveloped, with some topics mentioned without full explanation or appropriate citations. For example, the section on macrophage polarization in muscle regeneration (lines 358-369) is only supported by a few references, despite the existence of extensive studies by the Chazaud and Mounier groups that should be cited. Similarly, the discussion of satellite cell quiescence and activation linked to metabolic shifts is underdeveloped. Including the seminal works of Rodgers et al., Pala et al. (2018), Zhang et al. (2016), and L’honoré et al. (2018) would improve clarity.
3. The document strikes a balance between explaining theoretical aspects of autophagy and its practical applications in therapies for muscle-related diseases. However, in the section on the autophagy process, while there are a few diagrams, given the complexity of the processes described, more visual aids might help readers grasp the information more effectively. For example, Figure 1 should be moved from the Introduction to where the authors discuss each autophagy point to better engage readers.
4. The figures have inconsistent aesthetics. It is recommended that all figures be styled consistently. Furthermore, the quality of the figures does not meet publication standards, and there is a typo in Figure 2 ("differentitation" should be "differentiation").
5. A thorough review of all citations is necessary, as several sentences do not correspond to the correct references. For example, the sentence "…after denervation, downregulation of autophagic flux results in an increase in MuSCs, along with elevated expression of early activation markers such as PAX7 and MyoD" incorrectly refers to the work of Henze et al., which does not address autophagy in their denervation model.
6. Lines 51-54: The authors mention that the autophagy process is associated with the "onset and progression of various diseases, and therapeutic modulation of autophagy through pharmacological or molecular interventions has shown potential." It would be beneficial to include specific examples of diseases and corresponding therapeutic strategies to support this statement. The article lacks a critical discussion on the limitations of current research or potential challenges in applying findings to clinical settings.
7. While the English is generally acceptable, it could be improved, and there are a few scattered misspellings throughout the text.
In conclusion, while this review provides valuable insights into autophagy's role in muscle regeneration, enhancing structural cohesion, developing underexplored topics, improving visual aids and figure quality, correcting citation errors, providing specific examples of therapeutic modulation, and refining language will significantly strengthen the manuscript. Overall, the document presents solid research but could improve in terms of organization and accessibility to enhance its overall readability.
Author Response
Dear Reviewer,
Thank you for your review and valuable suggestions on our paper. Based on your feedback, we have revised the manuscript, and the specific changes are marked in red for your convenience.
Comments1:The review is structured in a way that lacks cohesion between paragraphs, resulting in a descriptive list rather than an integrated analysis of the various aspects of autophagy biology. While the sections are clearly titled, some transitions between topics, such as between general autophagy mechanisms and specific diseases, feel abrupt. A better flow or linking between sections could improve the reading experience and the authors should create more critical and speculative connections between these aspects.
Response 1:Thank you for your careful review and valuable comments on our manuscript. We have made the following revisions to improve the logical flow throughout the text:In the discussion on how molecular targets such as FOXO and mTOR regulate autophagy, we have incorporated the dual role of autophagy in tissue regeneration, highlighting its function both as a waste clearance mechanism and as a promoter of regeneration.Specifically, we have elaborated on the dynamic regulation of mTOR across different stages of regeneration. For example, in the initial phase, the suppression of mTOR promotes autophagy activation to remove damaged organelles and proteins. In contrast, during the later stages, mTOR activation supports cell growth and differentiation. This section emphasizes the balance autophagy maintains between recycling cellular components and promoting regeneration, establishing a coherent logical progression.(Page 4-5,line 85-136)
Comments 2:Certain parts of the review are underdeveloped, with some topics mentioned without full explanation or appropriate citations. For example, the section on macrophage polarization in muscle regeneration (lines 358-369) is only supported by a few references, despite the existence of extensive studies by the Chazaud and Mounier groups that should be cited. Similarly, the discussion of satellite cell quiescence and activation linked to metabolic shifts is underdeveloped. Including the seminal works of Rodgers et al., Pala et al. (2018), Zhang et al. (2016), and L’honoré et al. (2018) would improve clarity.
Response 2:Thank you very much for your thoughtful and constructive feedback. We have carefully reviewed your suggestions and made the following changes to address the concerns you raised: we have expanded the discussion on macrophage polarization and incorporated the seminal studies by Chazaud and Mounier. In response to your comment regarding the underdeveloped discussion of satellite cell quiescence and metabolic shifts, we have thoroughly revised this section. We have now included the seminal works of Rodgers et al., Pala et al. (2018), Zhang et al. (2016), and L’honoré et al. (2018). These references have enabled us to better illustrate the metabolic reprogramming that occurs during the transition from quiescence to activation, such as the shift from oxidative phosphorylation to glycolysis. Additionally, we now emphasize how ROS levels and autophagy influence this process, ensuring greater clarity and depth in our analysis.(Page12,15-16)
Comments 3:The document strikes a balance between explaining theoretical aspects of autophagy and its practical applications in therapies for muscle-related diseases. However, in the section on the autophagy process, while there are a few diagrams, given the complexity of the processes described, more visual aids might help readers grasp the information more effectively. For example, Figure 1 should be moved from the Introduction to where the authors discuss each autophagy point to better engage readers.
Response 3:Thank you very much for your thoughtful feedback. We have reviewed the structure of the manuscript and agree that additional visual aids will enhance readers' understanding of the complex autophagy process. To address this, we have added a new diagram(Figure 1) illustrating the regulation of autophagy by mTORC1.Following your suggestion, we have moved Figure 1(now as Figure 3) from the Introduction to the section on the regulation of satellite cells by autophagy. We believe this adjustment will provide a more logical flow and improve reader engagement.(Page 4-5,9,12)
Comments 4:The figures have inconsistent aesthetics. It is recommended that all figures be styled consistently. Furthermore, the quality of the figures does not meet publication standards, and there is a typo in Figure 2 ("differentitation" should be "differentiation").
Response 4:Thank you for your valuable feedback. We have revised all figures to ensure a consistent aesthetic throughout the manuscript, and we have improved their resolution to meet publication standards. Additionally, the typo in Figure 2 (“differentitation”) has been corrected to “differentiation.” We appreciate your attention to these details, which has helped us enhance the overall quality of the presentation.(Page 9)
Comments 5:A thorough review of all citations is necessary, as several sentences do not correspond to the correct references. For example, the sentence "…after denervation, downregulation of autophagic flux results in an increase in MuSCs, along with elevated expression of early activation markers such as PAX7 and MyoD" incorrectly refers to the work of Henze et al., which does not address autophagy in their denervation model.
Response 5:Thank you very much for your careful review and insightful feedback. We have thoroughly re-examined all citations to ensure that the references correspond accurately to the content. Specifically, we have corrected the misattribution regarding the role of autophagy after denervation, ensuring that Henze et al.'s study is appropriately referenced. Additionally, we reviewed other sections to prevent similar discrepancies and ensure the accuracy of all citations.(Page 12,line 262-267)
Comments 6:Lines 51-54: The authors mention that the autophagy process is associated with the "onset and progression of various diseases, and therapeutic modulation of autophagy through pharmacological or molecular interventions has shown potential." It would be beneficial to include specific examples of diseases and corresponding therapeutic strategies to support this statement. The article lacks a critical discussion on the limitations of current research or potential challenges in applying findings to clinical settings.
Response 6:Thank you very much for your valuable feedback. In response, we have revised the manuscript to include specific examples of therapeutic strategies related to autophagy, such as spermidine and pomegranate polyphenols, in the section on therapeutic directions. Additionally, we have discussed the limitations of current research and the challenges of clinical application.(Page 19)
Comments 7:While the English is generally acceptable, it could be improved, and there are a few scattered misspellings throughout the text.
Response 7:Thank you for your feedback. In response, we have thoroughly reviewed the manuscript to correct scattered misspellings and refine the language. We also improved the clarity, fluency, and overall readability of the text to ensure it meets the standards.
Sincerely,
Yun Chu
[Your Name]
Reviewer 3 Report
Comments and Suggestions for Authors
After reviewing the excellent review entitled "Autophagy in Muscle Regeneration: Mechanisms, Targets, and Therapeutic Perspective" by Chu et al. submitted for consideration as a potential publication in IJMS, as a reviewer I can state the following:
1. Overall, the paper is well written and describes in great detail the issues raised in the title of the manuscript.
2. The authors present a well-done review that makes understandable the different biochemical mechanisms of muscle autophagy in physiological and pathological conditions. In addition, the figures provided in the manuscript break the monotony of the manuscript and, in turn, enhance the reader's understanding of the mechanisms involved in this biological phenomenon.
3. As a reviewer and considering the topicality of the issue, I would consider it important for the authors to expand the information on the mechanisms of autophagy and consequently muscle regeneration in relation to fasting and muscle exercise at different intensities and how cyrcadian rhythms could affect this biological process.
Author Response
Dear Reviewer,
Thank you for your review and valuable suggestions on our paper.
Comments1:Overall, the paper is well written and describes in great detail the issues raised in the title of the manuscript.
Response 1:Thank you for your review and positive feedback on our paper, particularly for recognizing the structure and thoroughness of our work. We are pleased to hear that you found the paper well-aligned with the issues raised in the title. Following your feedback, we conducted a careful review to further refine the details, aiming to enhance the depth and breadth of our research presentation.
Comments 2:The authors present a well-done review that makes understandable the different biochemical mechanisms of muscle autophagy in physiological and pathological conditions. In addition, the figures provided in the manuscript break the monotony of the manuscript and, in turn, enhance the reader's understanding of the mechanisms involved in this biological phenomenon.
Response 2:Thank you for your high praise of our work.We are pleased that you found our review accessible and that it effectively explained the complexities of this topic. Additionally, we are glad that you felt the figures contributed to enhancing the reader’s understanding of this biological phenomenon. We placed great emphasis on integrating visual aids to provide a more intuitive grasp for readers, and we sincerely appreciate your positive feedback.
Comments 3:As a reviewer and considering the topicality of the issue, I would consider it important for the authors to expand the information on the mechanisms of autophagy and consequently muscle regeneration in relation to fasting and muscle exercise at different intensities and how cyrcadian rhythms could affect this biological process.
Response 3:Thank you for your insightful feedback and for emphasizing the importance of exploring the mechanisms of autophagy in relation to fasting, exercise intensity, and circadian rhythms. While we agree that these aspects are relevant, our review is specifically more focused on autophagy-related therapies and their future clinical applications. Expanding into the physiological mechanisms under various conditions would broaden the scope beyond our intended focus.
Sincerely,
Yun Chu